# Hidden magnetism uncovered in a charge ordered bilayer kagome material ScV₆Sn₆

Z. Guguchia [1] ✉, D. J. Gawryluk [2] ✉, S. Shin [2,12], Z. Hao[3,12], C. Mielke III [1,4], D. Das [1], I. Plokhikh[2], L. Liborio[5], J. Kane Shenton[5], Y. Hu[6], V. Sazgari [1], M. Medarde [7], H. Deng[3], Y. Cai[8], C. Chen [8], Y. Jiang [9], A. Amato [1], M. Shi [6], M. Z. Hasan [9,10,11], J.-X. Yin [3], R. Khasanov [1], E. Pomjakushina [2] & H. Luetkens [1]

Charge ordered kagome lattices have been demonstrated to be intriguing platforms for studying the intertwining of topology, correlation, and magnetism. The recently discovered charge ordered kagome material ScV₆Sn₆ does not feature a magnetic groundstate or excitations, thus it is often regarded as a conventional paramagnet. Here, using advanced muon-spin rotation spectroscopy, we uncover an unexpected hidden magnetism of the charge order. We observe an enhancement of the internal field width sensed by the muon ensemble, which takes place within the charge ordered state. More importantly, the muon spin relaxation rate below the charge ordering temperature is substantially enhanced by applying an external magnetic field. Taken together with the hidden magnetism found in $A$V₃Sb₅ ($A$ = K, Rb, Cs) and FeGe kagome systems, our results suggest ubiqitous time-reversal symmetry-breaking in charge ordered kagome lattices.

One of the most scientifically fruitful families of layered systems shown to host intriguing and exotic physics are the kagome lattice materials[1–5], composed of a two-dimensional lattice of corner-sharing triangles. This unique geometry gives rise to frustration, correlated quantum orders, and topology, owing to its special features: the electronic structure hosts a flat band, inflection points called "van Hove singularities", and Dirac cones.

The recently discovered $A$V₃Sb₅ ($A$ = K, Rb, Cs)[6–8] family of materials, crystallizing in a structurally perfect two-dimensional kagome net of vanadium atoms, shows the typical kagome band structure and exhibits two correlated orders: high-temperature charge order and a superconducting instability at lower temperatures[9–11]. The unique feature of $A$V₃Sb₅ is the emergence of a time-reversal symmetry (TRS) breaking chiral charge order with both magnetic[12–20] and electronic anomalies[21,22]. Theoretically, these features could be explained by a complex order parameter realizing a higher angular momentum state, dubbed unconventional, in analogy to superconducting orders[23–28].

$R$V₆Sn₆ ($R$ = Sc, Y, or rare earth)[29–32] is a more recent family of kagome metals that has a similar vanadium structural motif as the $A$V₃Sb₅ ($A$ = K, Rb, Cs) compounds. However, ScV₆Sn₆ is the only compound among the series of $R$V₆Sn₆ that has been found to host charge order below $T^* \simeq 90$ K[32]. Some distinct characteristics[33–36] of

[1]Laboratory for Muon Spin Spectroscopy, Paul Scherrer Institute, CH-5232 Villigen PSI, Switzerland. [2]Laboratory for Multiscale Materials Experiments, Paul Scherrer Institut, 5232 Villigen PSI, Switzerland. [3]Department of Physics, Southern University of Science and Technology, Shenzhen, Guangdong 518055, China. [4]Physik-Institut, Universität Zürich, Winterthurerstrasse 190, CH-8057 Zürich, Switzerland. [5]Scientific Computing Department, Science & Technology Facilities Council, Rutherford Appleton Laboratory, Didcot OX11 0QX, UK. [6]Photon Science Division, Paul Scherrer Institut, CH-5232 Villigen PSI, Switzerland. [7]Laboratory for Multiscale Materials Experiments, Paul Scherrer Institut, CH-5232 Villigen PSI, Switzerland. [8]Shenzhen Institute for Quantum Science and Engineering, Southern University of Science and Technology, Shenzhen 518055, China. [9]Laboratory for Topological Quantum Matter and Advanced Spectroscopy (B7), Department of Physics, Princeton University, Princeton, NJ 08544, USA. [10]Princeton Institute for the Science and Technology of Materials, Princeton University, Princeton, NJ 08540, USA. [11]Quantum Science Center, Oak Ridge, TN 37831, USA. [12]These authors contributed equally: S. Shin, Z. Hao. ✉e-mail: zurab.guguchia@psi.ch; dariusz.gawryluk@psi.ch

charge order such as different propagation vectors and distinct electron-phonon coupling strength[36] have been reported in these two families of kagome metals. However, $ScV_6Sn_6$ shows no superconductivity even after suppression of charge order by hydrostatic pressure[37]. One of the most intriguing questions about the charge order in $ScV_6Sn_6$ is whether or not it breaks time-reversal symmetry.

In this paper, we utilize the combination of zero-field (ZF) and transverse-field (TF) $\mu$SR techniques to probe the $\mu$SR relaxation rates in $ScV_6Sn_6$ as a function of temperature and magnetic field. We detect an enhancement of the muon-spin depolarization rate below the charge order temperature $T^*$. The rate is further and significantly enhanced by applying an external magnetic field along the crystallographic $c-$axis, which is indicative of a strong contribution of electronic origin to the muon spin relaxation below the charge ordering temperature. These results point to a complex electronic response strongly intertwined with the charge order in the kagome system $ScV_6Sn_6$ and provide useful insights into the microscopic mechanisms involved in the charge order.

## Results and discussion

The layered crystal structure of $ScV_6Sn_6$ with its V atoms arranged in a kagome network is shown in Fig. 1a. Figure 1b presents the temperature dependence of the specific heat capacity, $C_p$. A well pronounced peak in the heat capacity, corresponding to the charge order transition at $T^* \simeq 80$ K, is clearly seen, which provides evidence of a bulk phase transition. Single-crystal X-ray (see the Supplementary Note 1 and the Supplementary Table 1) diffraction measurements (see Fig. 1c–e) confirm that the as-grown sample crystallizes in the space group $P6/mmm$; the lattice parameters $a = 5.4739(3)$ Å and $c = 9.1988(7)$ Å at room temperature are close to those reported in previous studies[32]. Good agreement between the measured and calculated X-ray diffraction intensities (Fig. 1e) confirms that the sample is of high-quality. The deviations from the ideal occupancies on each atomic site are within $3\sigma$

and the difference Fourier maps show no additional features, indicating that the crystal structure is ordered and stoichiometric.

Magnetotransport data provide further evidence for the charge order transition. Magnetotransport has been shown to be particularly sensitive for detecting the charge order transition, since the magnetoresistance (MR) is a measure of the mean free path integrated over the Fermi surface and can detect a change of the scattering anisotropy and/or a Fermi surface reconstruction reliably. The MR under perpendicular magnetic field in $ScV_6Sn_6$ from 1.9 K to 120 K is shown in Fig. 2a. At high temperatures, MR adopts a standard quadratic dependence with $\mu_0H$ ($\mu_0H$ is the applied magnetic field) and can be well fitted to a polynomial: $\Delta\rho/\rho_{H=0} = \alpha + \beta(\mu_0H)^n$ (where $\alpha$, $\beta$, and $n$ are fitting parameters) with $n$ close to 2. MR shows deviation from quadratic field dependence upon lowering the temperature and becomes nearly linear deep within the charge-ordered state (see the Supplementary Notes 2, 3 and the Supplementary Figs. 1–4). The exponent is estimated to be $n \simeq 1.2$ at 1.9 K. Linear MR has usually been considered as a hallmark feature of an unconventional quantum state as such behavior was peviously observed in unconventional superconductors[38], topological[39], and charge/spin density wave materials[21,40]. For further insight, we provide the so called Kohler plot, $\Delta\rho/\rho_{H=0}$ vs $(\mu_0H/\rho_{H=0})^2$, which is shown in Fig. 2b. The most striking feature of this plot is that the MR data collapse to one line for all temperatures below $T^* \simeq 80$ K while they show a pronounced $T$-dependence above $T^*$. This effect may arise from a Fermi surface reconstruction caused by the charge order.

Our scanning tunneling microscopy (STM) experiments provide direct confirmation for the charge order with an in-plane unit cell enlargement of $\sqrt{3} \times \sqrt{3}$ (see Fig. 2c). The angle-resolved photoemission spectroscopy (ARPES) shows that the Fermi surface has stronger intensities at K and $K'$ points (see Fig. 2d and also ref. 36). The autocorrelation of such Fermi surface data is consistent with instabilities at vectors featuring a $\sqrt{3} \times \sqrt{3}$ unit cell enlargement.

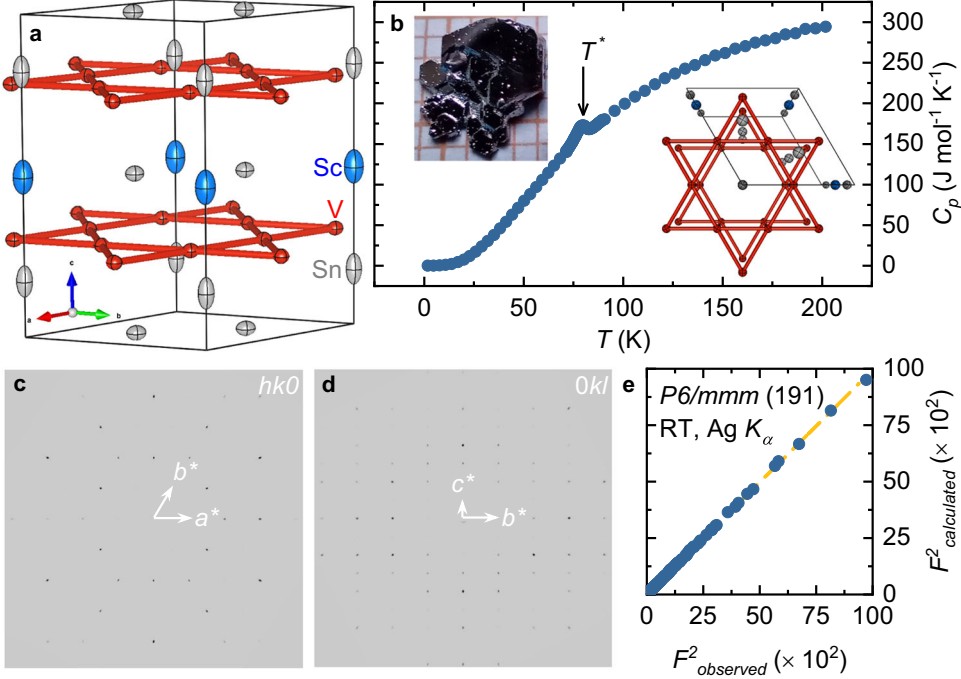

**Fig. 1 | Crystal structure of $ScV_6Sn_6$. a** Room temperature (RT) crystal structures of $ScV_6Sn_6$ with atomic displacement ellipsoids as determined from laboratory single-crystal X-ray diffraction (Ag $K_\alpha$) using the Space Group (SG) $P6/mmm$ (191) highlighting the kagome lattice pattern of the vanadium atoms. **b** Temperature dependence of the heat capacity for the $ScV_6Sn_6$ single crystals. Insets show a photograph of $ScV_6Sn_6$ single crystal and the crystal structure along c-direction.

**c, d** Reciprocal ($hk0$) and ($0kl$) respectively, lattice planes measured by laboratory X-ray diffraction at RT for a $ScV_6Sn_6$ single crystal, illustrating the agreement between the observed systematic absences and those of the $P6/mmm$ SG. **e** Agreement between the observed and calculated diffraction data at RT, as obtained from structural refinements (see Methods).

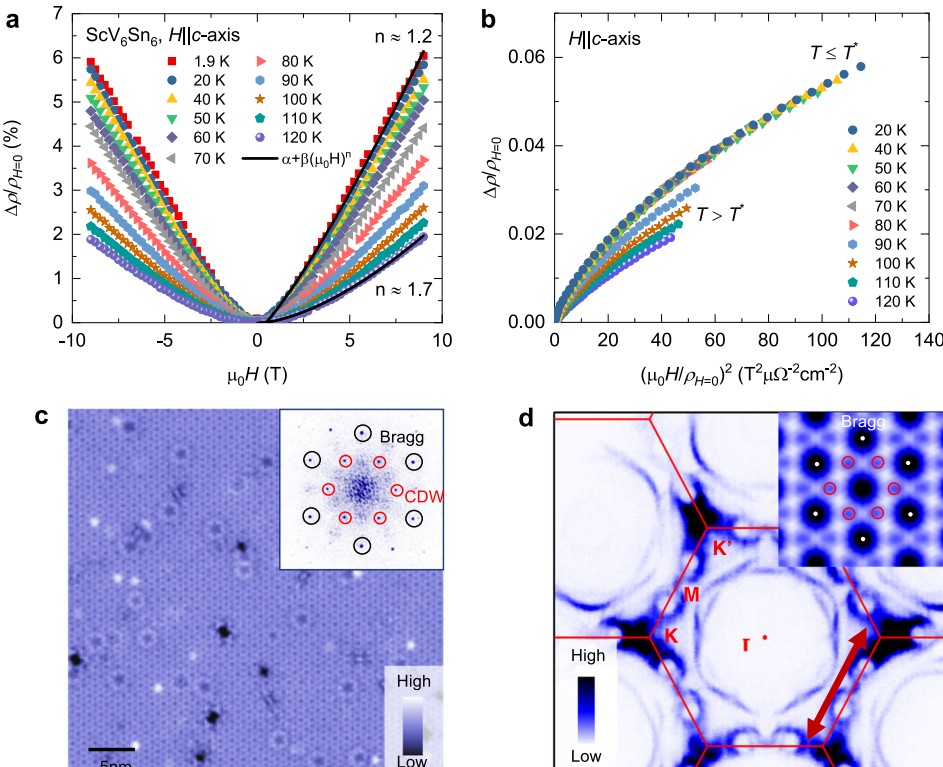

**Fig. 2 | Magnetotransport characteristics and charge order with an in-plane unit cell enlargement of $\sqrt{3} \times \sqrt{3}$ for ScV$_6$Sn$_6$. a** The magnetoresistance measured at various temperatures above and below the charge ordering temperature $T \simeq 80$ K. Black solid lines represent fits to the data by means of the following equation: $\Delta\rho/\rho_{H=0} = \alpha + \beta(\mu_0 H)^n$ **b** Kohler plot, $\Delta\rho/\rho_{H=0}$ vs $(\mu_0 H/\rho_{H=0})^2$, of the magnetoresistance, plotted from field-sweeps at various temperatures. **c** STM dI/dV spectroscopic map and corresponding Fourier transform (inset). The red circled marks the $\sqrt{3} \times \sqrt{3}$ charge order. Data taken at $V = 100$ meV, $I = 0.5$ nA, $T = 4.5$ K, $V_{mod} = 10$ meV. **d** Fermi surface obtained by ARPES showing stronger intensities near K points. The inset shows the autocorrelation of the Fermi surface, highlighting the instability with a vector of $\sqrt{3} \times \sqrt{3}$ as marked by the red circles. This vector corresponds to the scattering between states at K and $K'$ points as marked by the arrow in the main figure. Data taken at 10 K.

Next, in order to search for any magnetism (static or slowly fluctuating) associated with charge order in ScV$_6$Sn$_6$, zero-field $\mu$SR experiments have been carried out above and below the charge order temperature $T^*$. A schematic overview of the experimental setup with the muon spin forming 45° with respect to the $c$-axis of the crystal is shown in Fig. 3a. The sample was surrounded by four detectors: Forward (1), Backward (2), Up (3), and Down (4) (see the Supplementary Note 4 and the Supplementary Figure 5). Figure 3b displays the zero-field $\mu$SR spectra from detectors 3 and 4 collected over a wide temperature range. The ZF-$\mu$SR spectrum is characterized by a weak depolarization of the muon spin ensemble and shows no evidence of long-range ordered magnetism in ScV$_6$Sn$_6$. However, it shows that the muon spin relaxation has a clearly observable temperature dependence. Figure 3c displays the $\mu$SR spectra collected at 5 K in zero-field and at various external magnetic fields applied longitudinal to the muon spin polarization, $B_{LF} \simeq 1$-25 mT. Since the full polarization can be recovered by the application of a small external longitudinal magnetic field, $B_{LF} = 5$ mT, the relaxation is, therefore, due to spontaneous fields which are static on the microsecond timescale[41]. Similar to AV$_3$Sb$_5$, the zero-field $\mu$SR spectra for ScV$_6$Sn$_6$ were fitted using the gaussian Kubo-Toyabe depolarization function[42], multiplied by an additional exponential exp(-$\Gamma t$) term

$$P_{ZF}^{GKT}(t) = \left(\frac{1}{3} + \frac{2}{3}\left(1 - \Delta^2 t^2\right)\exp\left[-\frac{\Delta^2 t^2}{2}\right]\right)\exp(-\Gamma t) \quad (1)$$

where $\Delta/\gamma_\mu$ is the width of the local field distribution primarily due to the nuclear moments. However, this Gaussian component

may also include the field distribution at the muon site created by a dense network of weak electronic moments. $\gamma_\mu/2\pi = 135.5$ MHz/T is the muon gyromagnetic ratio. The deviation from a purely GKT like spectrum which is accounted for by the exponential term $\Gamma$ may e.g. originate from the entanglement of the muon with neighboring quadrupolar nuclei[43], modification of the nuclear positions around the muon due to charge order, or dilute electronic moments. $\Delta_{34}$ shows a non-monotonous temperature dependence; namely, a peak coinciding with the onset of the charge order, which decreases to a broad minimum before increasing again towards lower temperatures. $\Delta_{12}$ instead shows a weak minimum at $T^*$ with the significant increase at lower temperatures. The onset of charge order might alter the electric field gradient (EFG) experienced by the nuclei, due to the fact that the quantization axis for the nuclear moments depends on the electric field gradients, and correspondingly the magnetic dipolar coupling of the muon to the nuclei[44]. This can induce a change in the nuclear dipole contribution to the zero-field $\mu$SR signal and may explain the small maximum or minimum in $\Delta_{34}$ and $\Delta_{12}$, respectively, at the onset of $T^*$. However, the significant increase of both $\Delta_{34}$ and $\Delta_{12}$ at lower temperatures is difficult to explain with the change of the EFG and suggests a considerable contribution of electronic origin (dense moments) to the muon spin relaxation in the charge ordered state. There is also a noteworthy increase in the electronic relaxation rate $\Gamma_{34}$ upon lowering the temperature below the charge ordering temperature $T^*$, which is better visible in the inset of Fig. 3d. Another possible reason for the change of the relaxation rate across $T^*$ could be an

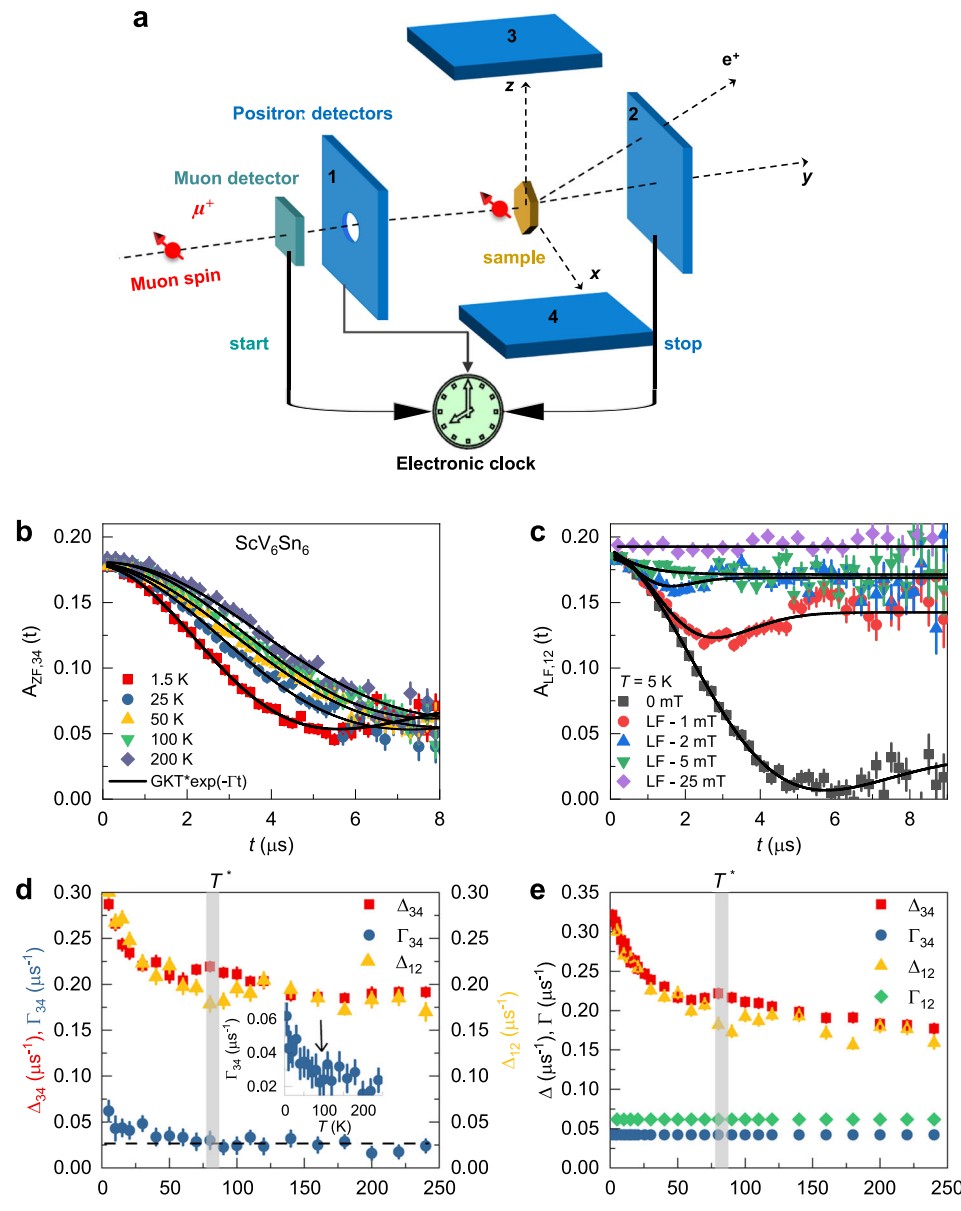

**Fig. 3 | Magnetic response of the charge order in ScV$_6$Sn$_6$. a** Overview of the experimental setup. Spin polarized muons with spin $S_\mu$, forming 45° with respect to the *c*-axis of the crystal, are implanted in the sample, which is surrounded by four positron detectors: Forward (1), Backward (2), Up (3), and Down (4). A clock is started at the time the muon surpasses the muon detector (M) and is stopped as soon as the decay positron is detected in one of the detectors. **b** The ZF $\mu$SR time spectra for ScV$_6$Sn$_6$, obtained at different temperatures. The solid black curves in panel **b** represent fits to the recorded time spectra, using the Eq. 1. Error bars are the standard error of the mean (s.e.m.) in about 10$^6$ events. **c** The $\mu$SR time spectra for ScV$_6$Sn$_6$, obtained at $T = 5$ K in zero-field and at various external magnetic field applied longitudinal to the muon spin polarization. The solid black curves in panel c represent fits to the recorded time spectra, using the Eq. 1. Error bars are the standard error of the mean (s.e.m.) in about 10$^6$ events. **d**, **e** The temperature dependences of the relaxation rates $\Delta$ and $\Gamma$ from two sets of detectors, obtained in a wide temperature range across the charge ordering temperature $T^* \simeq 80$ K. In panel **e** the rates $\Gamma_{34}$ and $\Gamma_{12}$ are kept constant as a function of temperature. The error bars represent the standard deviation of the fit parameters.

additional modulation of the lattice structure that slightly alters the nuclear positions around the muon[43]. However, a rough order of magnitude estimate yields that the structural distortions of the order of 0.1 Å for the atoms closest to the muon would be needed to explain the observed effect in the second moment of the measured field distribution. This is a large effect that has not been seen by any other technique. Moreover, this lattice distortion should be varying in temperature with a rather unusual trend. Therefore, we can dismiss the structural distortion being origin for the increase of the relaxation rate. Most importantly, our high field $\mu$SR results presented below definitively prove that

there is indeed a strong contribution of electronic origin to the muon spin relaxation below the charge ordering temperature (see below). In sum, the ZF-$\mu$SR results indicate that there is an enhanced width of internal fields sensed by the muon ensemble below $T^* \simeq 80$ K. The increase of $\Delta$ and $\Gamma$ below $T^*$ as well as the dip-like feature around $T^*$ are reminiscent of the behavior observed in other kagome systems AV$_3$Sb$_5$ (A = K, Rb, Cs). The increase in $\Delta_{34}$ and $\Delta_{12}$ below $T^*$ in ScV$_6$Sn$_6$ is estimated to be $\simeq 0.1$ $\mu$s$^{-1}$ and $\simeq 0.12$ $\mu$s$^{-1}$, respectively, which can be interpreted as a characteristic field strength $\Delta_{34,12}/\gamma_\mu \simeq 1.2$–1.5 G. Since we see enhanced electronic relaxation below $T^*$ in both $\Delta_{12}$ and $\Delta_{34}$, we

conclude that the local field cannot lie purely along the $c$-axis direction (this would lead to an absence of the terms $\lambda_{34}$ and $\Delta_{34}$). However, any orientation of the local field which has a significant component in the basal plane is consistent with our data. We note that the value of the internal field, obtained for $ScV_6Sn_6$ is by factor of ~4–5 higher than the one reported previously for $AV_3Sb_5$ (A = K, Rb, Cs). A similar increase of internal magnetic field strength is reported in several time-reversal symmetry-breaking superconductors[45] and in some multigap TRS breaking superconductors (e.g. $La_7Ni_3$[46]) across $T_c$.

In order to substantiate the zero-field $\mu SR$ results, presented above, we carried out systematic high field $\mu SR$[47] experiments. Under high magnetic field, the direction of the applied field defines the quantization axis for the nuclear moments, so that the effect of the charge order on the electric field gradient at the nuclear sites is irrelevant. For the high-field experiments, a mosaic of several crystals was used. The individual crystals were glued to a 10 mm circular silver sample holder and the entire ensemble was held together by small droplets of GE varnish. Figure 4a shows the probability field distribution, measured at 3 K in a magnetic field of 8 T applied along the crystallographic $c$-axis. In the whole investigated temperature range, two-component signals were observed: a signal with fast relaxation $\sigma_{TF} \simeq 0.428(3)$ $\mu s^{-1}$ (broad signal on the left side of the Fourier spectrum) and another one with a slow relaxation $0.05$ $\mu s^{-1}$ (narrow signal). The narrow signal arises mostly from the muons stopping in the silver sample holder and its position is a precise measure of the value of the applied magnetic field. The width and the position of the narrow signal is found to be temperature independent, as expected, and thus were kept constant in the analysis (see the Supplementary Note 5 and the Supplementary Fig. 6). The relative fraction of the muons stopping in the sample was fixed to the value obtained at the base-$T$ and kept temperature independent. The signal with the fast relaxation, which is shifted towards a lower field from the applied one, arises from the muons stopping in the sample and it takes a major fraction (~50 %) of the $\mu SR$ signal. This points to the fact that the sample response arises from the bulk of the sample. A non-monotonous behavior of the relaxation rate with two characteristic temperatures 25 K and $T^* = 80$ K is clearly seen in the $\mu SR$ data, measured in magnetic field of 0.01T, applied parallel to the $c$-axis, as shown in Fig. 4b. The Knight shift measurements (see the Supplementary Note 6 and the Supplementary Fig. 7) also indicate two temperature scales, showing a decrease at the

charge order transition temperature $T^* = 80$ K followed by an increase at ~25 K. The non-monotonous behavior of the rate in 0.01T looks similar to the temperature dependence of the zero-field relaxation rate $\Delta_{34}$. This shows that the effect of nuclear contribution on the relaxation rate at the charge order transition is still visible in low fields. However, at the field of 2T the rate does not show decrease of the rate below $T^*$ but rather shows a clear and monotonous increase with decreasing temperature with the onset of $T^*$. At higher fields such as 4T, 6T, and 8T, the rate shows stronger increase towards low temperatures within the charge ordered state. As the nuclear contribution to the relaxation cannot be enhanced by an external field, this indicates that the low-temperature relaxation rate in magnetic fields is dominated by the electronic contribution and minor effect of nuclear contribution at the transition is diminished. Remarkably, we find that the absolute increase of the relaxation rate between the onset of charge order $T^*$ and the base-$T$ in 8 T is $\Delta\sigma_{TF} \simeq 0.23$ $\mu s^{-1}$ which is a factor of two higher than the one $0.12$ $\mu s^{-1}$ observed in zero-field. The transverse-field relaxation is normally smaller by a factor of ~ 2 than expected from the zero-field measurements[41]. This is because in transverse-field measurements only the width in one direction is relevant while in ZF the width in two directions is relevant. This means that the relaxation for $ScV_6Sn_6$ in high transverse-field is actually a factor of four higher than in ZF. This indicates a strong field-induced enhancement of the electronic response. We also note that a weak but non-negligible field effect on the relaxation rate is observed above $T^*$ within a 20–30 K temperature range. This may point towards the short-range charge order preceding the phase transition and becomes observable in $\mu SR$ under fields higher than 2 T. Short-range charge order was also suggested by inelastic X-ray scattering and magneto-transport measurements[48,49].

Based on the unique combination of ZF-$\mu SR$ and high-field $\mu SR$ experiments, we observed a magnetic response (enhanced internal field width) below $T^* \simeq 80$ K, providing direct evidence for the time-reversal symmetry-breaking fields in the kagome lattice of $ScV_6Sn_6$. Since at least 50% of the sample volume experiences an increase in the relaxation rate, this indicates the bulk nature of the transition below $T^*$. We note that Zero-field and low field relaxation rate as well as Knight-shift measurements indicate two characteristic temperatures, which may point towards two different order parameters. Our $\mu SR$ observation of time-reversal symmetry-breaking charge order in $ScV_6Sn_6$ is consistent with the recently reported transport measurements,

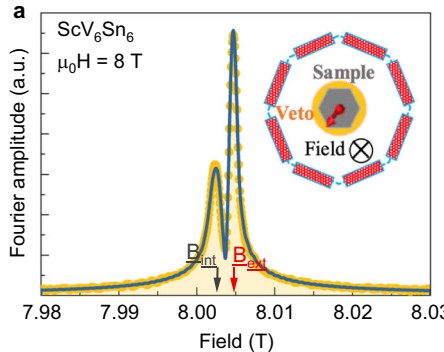

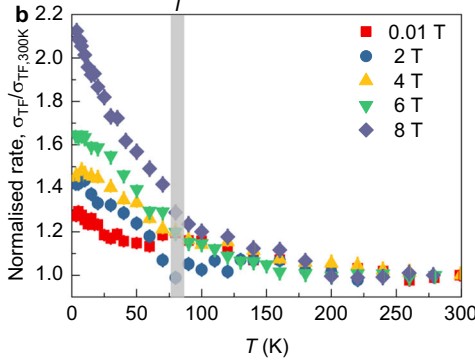

**Fig. 4 | Enhanced magnetic response of the charge order with applying external magnetic fields. a** Fourier transform of the $\mu SR$ asymmetry spectra for a mosaic of single crystals of $ScV_6Sn_6$ at 3 K in the presence of an applied field of $\mu_0 H = 8$T. The blue solid line is a two-component signal fit. The peaks marked by the arrows denote the external and internal fields, determined as the mean values of the field distribution from the silver sample holder and from the sample, respectively. The short-time-window apodization function was used in the Fourier transform amplitude plot. Inset shows the schematic high-field $\mu SR$ experimental setup. The sample was surrounded by 2 × 8 positron detectors, arranged in rings. The

specimen was mounted in a He gas-flow cryostat with the largest face perpendicular to the muon beam direction, along which the external field was applied. Behind the sample lies a veto counter (in orange) which rejects the muons that do not hit the sample. **b** Temperature dependence of the high transverse field muon spin relaxation rate $\sigma_{TF}$ for the single crystal of $ScV_6Sn_6$, normalized to the value at 300 K, measured under different $c$-axis magnetic fields. The vertical gray lines mark the charge ordering temperature, determined from specific heat measurements. The error bars represent the standard deviation of the fit parameters.

revealing the anomalous Hall effect below $T^{*}$[50,51]. We note that the presence of an electronic response and its field induced enhancement in $ScV_6Sn_6$ is similar to our previous observations in kagome lattice superconductors $AV_3Sb_5$ ($A$ = K, Rb, Cs)[12–14].

We specify some similarities and differences between $ScV_6Sn_6$ and $AV_3Sb_5$: (1) In $ScV_6Sn_6$, we observed the enhancement of the rate $\Gamma_{34}$ by 0.03 $\mu s^{-1}$ below $T^{*}$, which is similar to $KV_3Sb_5$. On the other hand, the Gaussian rate $\Delta_{34}$ and $\Delta_{12}$ show significant increase, i.e. by 0.12 $\mu s^{-1}$, in $ScV_6Sn_6$, while it shows only a very weak temperature dependence at low temperatures in $AV_3Sb_5$. These results show that the increase of relaxation in $ScV_6Sn_6$ is due predominantly to Gaussian rate. This indicates that internal fields arise from a network of dense weak electronic moments in $ScV_6Sn_6$, while the electronic moments seem to be more dilute or more dynamic in $AV_3Sb_5$. (2) Among the three compounds $AV_3Sb_5$ ($A$ = K, Rb, Cs) the largest increase of the zero-field relaxation rate below $T^{*}$ was observed for $RbV_3Sb_5$. In $ScV_6Sn_6$, the increase of the zero-field relaxation is 0.12 $\mu s^{-1}$, which is a factor of two higher than the one 0.06 $\mu s^{-1}$ observed in $RbV_3Sb_5$. (3) The largest field effect was observed in $KV_3Sb_5$. In $ScV_6Sn_6$, the magnetic field also leads to the enhancement of the rate and the increase of the rate reaches a value of 0.23 $\mu s^{-1}$ at 8 T. This is a factor of 1.6 higher than the one (0.15 $\mu s^{-1}$) observed in $KV_3Sb_5$. The $AV_3Sb_5$ systems are close to the condition of a kagome lattice with van Hove filling and with extended Coulomb interactions[12,23]. TRS breaking charge order in $AV_3Sb_5$ was interpreted in terms of orbital current order[23–28], which may fundamentally affect the superconducting state. Drastic magnetic-field-induced chiral current order was also reported theoretically[52] which lines up with our high field $\mu$SR results. Unconventional charge order and orbital current order has also been proposed for $ScV_6Sn_6$[53,54] based on Ginzburg-Landau and mean-field analysis. According to the theoretical modeling, there is very small net flux and thus the small net magnetic moment in the unit cell of the order. The suggested orbital current was reported to be homogeneous on the lattice, however alternating in its flow, which would produce inhomogeneous fields at the muon site. Within this framework, muons may couple to the closed current orbits below $T^{*}$, leading to an enhanced internal field width sensed by the muon ensemble concurrent with the charge order[12,13]. Despite the fact that $\mu$SR results seems to be compatible with the picture of orbital current order, we cannot conclude on the microscopic origin of the TRS breaking field in this system. But, our results provide key evidence that the magnetic and charge channels of $ScV_6Sn_6$ are strongly intertwined, which can give rise to complex and collective phenomena. This is an experimental finding which stands even without the knowledge of its microscopic origin. This will also inspire future experiments, particularly neutron scattering with polarization analysis, to potentially understand the precise origin of the observed magnetism.

The exploration of unconventional electronic phases that result from strong electronic correlations is a frontier in condensed matter physics. Kagome lattice systems appear to be an ideal setting in which strongly correlated topological electronic states may emerge. Most recently, the kagome lattice of $AV_3Sb_5$ has shown to host unconventional chiral charge order, which is analogous to the long-sought-after quantum order in the Haldane model[55] for honeycomb lattice or Varma model[56] for cuprate high-temperature superconductors. Here we employ muon spin relaxation to probe the magnetic response of kagome charge order on a microscopic level in newly discovered $ScV_6Sn_6$ with a vanadium kagome lattice. We found an enhancement of the internal field width sensed by the muon ensemble, which takes place within the charge ordered state. The muon spin relaxation rate below the charge ordering temperature is substantially enhanced by applying an external magnetic field. Our work points to a time-reversal symmetry-breaking charge order in $ScV_6Sn_6$ and extends the classification of materials with unconventional charge order beyond the series of compounds $AV_3Sb_5$ ($A$ = K, Rb, Cs).

## Methods

### Sample preparation

Single crystals of $ScV_6Sn_6$ were grown by the molten Sn flux method. The starting materials were Sc (3N, Alfa Aesar), V (2N8, Alfa Aesar), and Sn (99.9999%, Alfa Aesar). In a helium-filled glovebox, Sc (0.218 g, 0.485 mmol), V (1.481 g, 2.908 mmol), and Sn (33.390 g, 28.127 mmol) were placed into a 5 ml alumina Canfield crucible[57]. The crucible set was placed into a quartz u-tube, evacuated, back-filled with c.a. 100 mbar of Ar, and sealed as an ampoule. The specimen was heated up to 1150 °C, with a rate of 400 °C/h, and annealed at that temperature for 12 h. Subsequently, the sample was cooled down to 780 °C with a rate of 100 °C/h. After the growth step, the excess Sn flux was separated from the single crystals by centrifugation.

### Crystallography

A single crystal of $ScV_6Sn_6$, obtained from Sn flux, was mounted on the *MiTeGenMicroMounts* loop and used for a X-ray structure determination. Measurements were performed at RT on a *STOESTADIVARI* diffractometer equipped with a *DectrisEIGER1M2RCdTe* detector and with an *AntonPaarPrimux* 50 Ag/Mo dual-source using Ag $K_\alpha$ radiation ($\lambda$ = 0.56083 Å) from a micro-focus X-ray source and coupled with an *OxfordInstrumentsCryostream* 800 jet. The unit cell constants and an orientation matrix for data collection were obtained from a least-squares refinement of the setting angles of 16647 reflections in the range 6.7° < 2$\theta$ < 66.6°. A total of 3448 frames were collected using $\omega$ scans, 5 s exposure time and a rotation angle of 0.5° per frame, and a crystal-to-detector distance of 60.0 mm.

Data reduction was performed with *X-Area* package [*X-Area* package, Version 2.1, STOE and Cie GmbH, Darmstadt, Germany, 2022]. The intensities were corrected for Lorentz and polarization effects, and an empirical absorption correction using spherical harmonics was applied [X-Area package, Version 2.1, STOE and Cie GmbH, Darmstadt, Germany, 2022]. The structure was solved using *ShelXT*[58] and *Olexs*² program[59]. The model was refined with *ShelXL* package[60] and *Olexs*² software. Structures were plotted using the VESTA visualization tool[61].

### Heat capacity

Heat Capacity ($C_p$) measurements were performed in a Physical Properties Measurement System (PPMS, 14T, Quantum Design) at zero magnetic field using the relaxation method between 1.85 and 200 K. 7.75 mg of $ScV_6Sn_6$ crystal was fixed with the Apiezon-N grease to the sapphire holder of the calorimeter. The samples were then cooled down to 1.85 K, and the heat capacity measured by heating. The signal from the Apiezon-N grease and the calorimeter, previously measured under the same conditions, was subtracted from the data to obtain the sample's $C_p$.

### Magnetotransport

Magnetoresistance of the single-crystal $ScV_6Sn_6$ was measured in physical property measurement system (PPMS-9, Quantum Design) using the typical four-probe technique. Four Pt-wires (0.0254 mm diameter) were attached by silver epoxy on the single crystal, polished into bar-shaped specimen. Electrical current of 1 mA and magnetic field was applied along the crystallographic $a$- and $c$-axis, respectively, that were confirmed by Laue measurement.

### $\mu$SR experiment

In a $\mu$SR experiment nearly 100% spin-polarized muons $\mu^{+}$ are implanted into the sample one at a time. The positively charged $\mu^{+}$ thermalize at interstitial lattice sites, where they act as magnetic microprobes. In a magnetic material the muon spin precesses in the local field $B_\mu$ at the muon site with the Larmor frequency $\nu_\mu = \gamma_\mu/(2\pi)B_\mu$ (muon gyromagnetic ratio $\gamma_\mu/(2\pi) = 135.5$ MHz T$^{-1}$).

Zero field (ZF) and transverse field (TF) $\mu$SR experiments on the single crystalline sample of $ScV_6Sn_6$ were performed on the GPS instrument and high-field HAL-9500 instrument, equipped with Blue-Fors vacuum-loaded cryogen-free dilution refrigerator (DR), at the Swiss Muon Source (S$\mu$S) at the Paul Scherrer Institut, in Villigen, Switzerland. A mosaic of several crystals stacked on top of each other was used for these measurements. The magnetic field was applied along the crystallographic $c$-axis. A schematic overview of the experimental setup for zero-field is shown in Fig. 4a. The crystal was mounted such that the $c$-axis of it is parallel to the muon beam. Using the "spin rotator" at the $\pi$M3 beamline, muon spin was rotated (from its natural orientation, which is antiparallel to the momentum of the muon) by 44.5(3)° degrees with respect to the $c$-axis of the crystal. So, the sample orientation is fixed but the muon spin was rotated. The rotation angle can be precisely estimated to be 44.5(3)° by measurements in weak magnetic field, applied transverse to the muon spin polarization. Zero field and high transverse field $\mu$SR data analysis on single crystals of $ScV_6Sn_6$ were performed using both the so-called asymmetry and single-histogram modes[41,62]. The experimental data were analyzed using the MUSRFIT package[62].

## Data availability

All relevant data are available from the authors. Alternatively, the data can be accessed through the data base at the following link http://musruser.psi.ch/cgi-bin/SearchDB.cgi using the following details: 1. Area: GPS. Year: 2022. Run Title: $ScV_6Sn_6$. Run from 3393 to 3918. 2. Area: HAL. Year: 2022. Run Title: $ScV_6Sn_6$. Run from 1342 to 1484.

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

## Acknowledgements

The $\mu$SR experiments were carried out at the Swiss Muon Source (S$\mu$S) Paul Scherrer Insitute, Villigen, Switzerland. Z.G. acknowledges support from the Swiss National Science Foundation (SNSF) through SNSF Starting Grant (No. TMSGI2_211750). Z.G. acknowledges the fruitful discussions with Dr. Robert Scheuermann. Y.H. and M.S. acknowledges support from the Swiss National Science Foundation under Grant. No. 200021_188413. S.S. and E.P. acknowledge support from the Swiss National Science Foundation under Grant. No. 200021_188706. I.P. acknowledges support from Paul Scherrer Institute research grant No. 2021_01346. J.X.Y. acknowledges the support from the National Science Foundation of China (NSFC) (No. 12374060). M.Z.H. acknowledges support from the US Department of Energy, Office of Science, National Quantum Information Science Research Centers, Quantum Science Center at ORNL and Laboratory for Topological Quantum Matter at Princeton University.

## Author contributions

Z.G. conceived and designed the project. Z.G. and D.J.G. supervised the project. Sample growth, single crystal X-ray diffraction experiments and corresponding discussions: D.J.G., I.P., S.S., Z.H., Z.G., and E.P.; Transport experiments: S.S. in consultation with M.M. and Z.G.; $\mu$SR experiments and corresponding discussions: Z.G., C.M.III, D.D., V.S., L.L., J.K.S. A.A., R.K., and H.L.; STM and ARPES experiments and corresponding discussions: J.-X.Y., Y.J., H.D., Y.C., C.C., Y.H., M.S., D.J.G., Z.G., and M.Z.H.; Figure development and writing the paper: Z.G. with contributions from D.J.G. and all authors. All authors discussed the results, interpretation, and conclusion.

## Competing interests

The authors declare no competing interests.
