## [Peer Review File · Nature Communications]

REVIEWER COMMENTS

Reviewer #1 (Remarks to the Author):

The authors of the manuscript perform an experimental study of the kagome bilayer compound ScV_6Sn_6 , for which a charge-ordered state at a transition temperature similar to the one of the single layer kagome metals AV_3Sb_5 ($A = \text{K, Rb, Cs}$) was found. They perform a complete characterization of the material employing several techniques, ranging from magnetoresistance, scanning tunneling microscopy, angle-resolved photoemission spectroscopy and muon-spin rotation spectroscopy both in zero- and transverse-fields setups. With the latter technique, they find broken time-reversal symmetry in the charge-ordered state observed below 80K. This result is not in contradiction with the ones of other experiments uploaded on arXiv after the present work, where an anomalous Hall effect has been claimed [Mozaffari et al., arXiv:2305.02393, Yi et al., arXiv:2305.04683].

The experimental findings presented in this manuscript will be of certain interest to the community working in this field, and they seem to combine even more the physics of AV_3Sb_5 ($A = \text{K, Rb, Cs}$) with the one of ScV_6Sn_6 , despite the fact that the ordering wave vector for the charge order is different in the two cases. I believe these results will trigger additional experimental and theoretical works trying to further explore the similarities and differences among these compounds. Moreover, the article is presented clearly with figures which are, in most cases, easy to understand. Considering these factors, I hope the paper can be accepted for publication if the authors can make the following revisions.

1) In the manuscript, the acronym CDW is used, but it is never defined. If its meaning is the standard one, i.e., charge density wave, I would be more cautious regarding its use. Indeed, the debate concerning the nature of the charge order in the kagome metals seems far to be settled to me. Thus, I would only talk about charge order without further specifications, as the authors do in most of the manuscript.

2) Can the authors provide, perhaps in the Supplemental Material, the plot of the fitting parameters vs temperature for the curves displayed in Fig.2a, i.e., α , β and n ? This might simplify the reproducibility of the figure. Moreover, the authors should make consistent the formulas for the fit reported in the legend of Fig.2a with the ones reported in the caption of Fig.2 and at the end of the first column of the second page of the manuscript, where a factor μ_0 is missing.

3) Concerning Fig.2d, the color bar appearing there has to be corrected. Moreover, there are some features of the ARPES map that appear (or, at least, are clearly visible) just in the upper half of the first Brillouin zone, especially in the region closer to the Gamma point. Can the authors explain the origin of this asymmetry? Can it be regarded as a signature of nematicity?

4) At the end of the first column of page 6 of the manuscript, the authors state: “So far, no theoretical proposal for the orbital current order was reported for ScV₆Sn₆.” However, I am not sure this is the case. Indeed, in [Grandi et al., Phys. Rev. B 107, 155131 (2023)], a state that hosts orbital currents is also suggested for the $\sqrt{3}\times\sqrt{3}$ reconstruction of the kagome lattice, even if only a phenomenological description of the state is provided. As a consequence, I think the quoted sentence has to be modified.

5) I suggest you to describe, maybe in the supplemental material, how the fit shown in Fig.4a is performed for the sake of reproducibility.

6) If I understand Fig.4b correctly, the error bars have almost the same size as the symbols. This makes them difficult to be seen, and for this reason, I suggest reducing the size of the symbols (same in Fig.3e).

Reviewer #2 (Remarks to the Author):

Kagome physics is in the frontier of condensed matter research. The authors found enhanced magnetic response in single crystal ScV₆Sn₆ below the charge order temperature $T^*\sim 80$ K using ZF- μ SR and high-field μ SR measurements, indicating the time-reversal symmetry-breaking in kagome paramagnet ScV₆Sn₆. However, I cannot recommend its publication in Nature Communications at the present stage, there are some comments on this work need to be addressed.

1 The specific heat capacity curve shows a peak around 80 K, indicating the charge order transition. Is there the same response in the magnetic susceptibility and resistivity curves? Moreover, the transition temperature is lower than that ~ 90 K in previous work, is this related to the quality of sample?

2 The data in Fig.3 (d)-(e) are obtained from two sets of detectors, can they describe more in detail? It is noticed that the rate Γ_{34} increases about $0.03 \mu\text{s}^{-1}$ below T^* in Fig.3 (d), but it shows almost temperature independent in the whole temperature range in Fig.3 (e).

3 The relaxation rate under high fields shows clear increase towards low temperatures. Around T^* , it shows different behavior (non-monotonous or monotonous) under various fields, can they explain it?

4 The authors compared some similarities and differences between ScV6Sn6 and AV3Sb5, what is the novelty of this work?

5 There are some errors in the manuscript. For example, the labels (b)-(d) in Fig.3 are unclear. In addition, the color of the fitting line in Fig.4 (a) is blue, not black.

Reviewer #3 (Remarks to the Author):

The manuscript reports on very interesting muon spin spectroscopy

findings in a new Kagome compound with striking similarities to those obtained by some of the authors on the very actively studied AV3Sb5 (135) series. It is very clearly written and it deserves consideration for Nature Communications in view of the many order parameters that these structures may host, and of their possible interplay. The magnetic nature of the low temperature muon relaxation is proven beyond doubt by the zero field (ZF), longitudinal field (LF) and high transverse field

(HTF) data of Fig. 3 and 4. However a number of points related to data interpretation need addressing:

1) An anomaly appears in the ZF Kubo-Toyabe (KT) static widths at the known onset of the CDW. This is attributed to the change in the EFG across the onset. However, a change in EFG could give rise to an anisotropic (almost) step-wise change in the relaxation, reflecting the temperature dependence of the CDW order parameter, i.e. widening towards low temperatures. The bump showing up in the 34 and not in the 12 pair of detectors is hard to reconcile with this interpretation, since it completely disappears below 50 K where the two relaxation rates overlap

again.

2) "At higher fields such as 2 T, 4 T, 6 T, and 8 T, the rate shows a clear and stronger increase towards low temperatures within the charge ordered state."

This statement implies that the magnetic origin of the rate correlates with T^* , but the correlation is much weaker than implied. In particular HTF data of Fig. 4b show a smooth continuous decrease across T^* . It seems compatible with a slowing down mechanism connected to a paramagnetic susceptibility of Curie-Weiss type, i.e. with a finite exchange interaction. This would be still non

trivial in a metal with CDW instabilities, but it must not necessarily imply orbital currents developing below the CDW transition.

3) "We also note that a weak but non-negligible field effect on the relaxation rate is observed above T^* within a 20-30 K temperature range, which may point towards the charge density wave fluctuations preceding the phase transition."

CDW fluctuations are very unlikely the cause of a relaxation that

seamlessly reconnects to magnetic fluctuations to produce a continuous rate variation across T^* . It is suggested that this mention is either expanded and justified fully, or dropped.

4) There is no reference in the text to the temperature dependence of the shift for the sample component of the HTF data. It would be interesting to compare this shift with the spin susceptibility, with temperature as an implicit parameter (Clogston-Jaccarino plot). The hyperfine coupling could be obtained this way and it could provide further insight. In any case, shifts and susceptibility are mandatory (in the main text or as supplemental information). They clarify whether the sample displays any macroscopic magnetic behavior at all, and whether magnetic impurities are present in the sample.

5) "The exponential term Γ may e.g. be due to the presence of electric field gradients, causing deviations from the GKT-like spectrum or dilute electronic moments. [...] There is also a note-worthy increase in the relaxation rate Γ 34 upon lowering the temperature below the charge ordering temperature T^* ...".

The Γ parameter belongs to a factor of the model function(1), i.e. it is assumed to originate from an uncorrelated mechanism. If the rate were temperature-independent it could account for an indirect EFG effect. However, the low temperature increase is irreconcilable with a CDW order parameter variation (even less, fluctuation), so the EFG explanation is shaky.

6) While the investigation of the magnetic properties of these compounds is of utmost importance, the comparison with the 135 materials where "Muons couple to the closed current orbits below T^* , leading to an enhanced internal field width sensed by the muon ensemble concurrent with the charge order" is less evident from the experimental data presented in the manuscript.

This has been partially discussed in the previous points, but it is also visible in the different trends of the relaxation rate observed in ZF and applied field that are seemingly uncorrelated with T^* (actually this is the case also for the ZF measurements of the 135, where an upturn in the relaxation rate happens below the CDW transition). In present case however one observes temperature dependent phenomena starting well below ($\sim 50\text{K}$ in ZF) and well above ($> 100\text{K}$ for TF $> 4\text{T}$) the critical

temperature of the CDW. Can the author comment on this point and provide further evidence of their statement?

Minor point:

Fig 4a caption does not specify the short-time-window apodization function used in the FT amplitude plot, and indeed adding a longer time-window function FT plot could show the narrow nature of the high frequency peak.

Typos:

* TRS is not defined.

* Correct the angel in "MuSR methods".

Reviewer #4 (Remarks to the Author):

“Hidden magnetism uncovered in charge ordered bilayer kagome material ScV_6Sn_6 (NCOMMS-23-18321-T)”.

Reply to the Reviewer 1:

1.1 Reviewer’s comment: *The authors of the manuscript perform an experimental study of the kagome bilayer compound ScV_6Sn_6 , for which a charge-ordered state at a transition temperature similar to the one of the single layer kagome metals AV_3Sb_5 ($A = \text{K}, \text{Rb}, \text{Cs}$) was found. They perform a complete characterization of the material employing several techniques, ranging from magnetoresistance, scanning tunneling microscopy, angle-resolved photoemission spectroscopy and muon-spin rotation spectroscopy both in zero- and transverse-fields setups. With the latter technique, they find broken time-reversal symmetry in the charge-ordered state observed below 80K. This result is not in contradiction with the ones of other experiments uploaded on arXiv after the present work, where an anomalous Hall effect has been claimed [Mozaffari et al., arXiv:2305.02393, Yi et al., arXiv:2305.04683]. The experimental findings presented in this manuscript will be of certain interest to the community working in this field, and they seem to combine even more the physics of AV_3Sb_5 ($A = \text{K}, \text{Rb}, \text{Cs}$) with the one of ScV_6Sn_6 , despite the fact that the ordering wave vector for the charge order is different in the two cases. I believe these results will trigger additional experimental and theoretical works trying to further explore the similarities and differences among these compounds. Moreover, the article is presented clearly with figures which are, in most cases, easy to understand. Considering these factors, I hope the paper can be accepted for publication if the authors can make the following revisions.*

Our response: We are very thankful to the reviewer for supporting our manuscript for publication in Nature Communications. We are delighted to find the reviewer’s appreciation of our work. Very interesting preprints, mentioned by the reviewer, has been added in the Reference list of the revised manuscript.

1.2 Reviewer’s comment: *In the manuscript, the acronym CDW is used, but it is never defined. If its meaning is the standard one, i.e., charge density wave, I would be more cautious regarding its use. Indeed, the debate concerning the nature of the charge order in the kagome metals seems far to be settled to me. Thus, I would only talk about charge order without further specifications, as the authors do in most of the manuscript.*

Our response: We agree with the reviewer on this point and refer to the “charge order” throughout the manuscript.

1.3 Reviewer’s comment: *Can the authors provide, perhaps in the Supplemental Material, the plot of the fitting parameters vs temperature for the curves displayed in Fig.2a, i.e., alpha, beta and n? This might simplify the reproducibility of the figure. Moreover, the authors should make consistent the formulas for the fit reported in the legend of Fig.2a with the ones reported in the caption of Fig.2 and at the end of the first column of the second page of the manuscript, where a factor μ_0 is missing.*

Our response: Figure R1 depicts the temperature dependences of the fitting parameters α , β , n . All three parameters show the change in the slope across the charge order transition temperature $T^* \approx 80$ K. Following the suggestion of the Reviewer, we show this figure in the supplementary information of the revised manuscript. We also added the missing factor μ_0 in the polynomial function.

Figure R1: The temperature dependences of the parameters n , α and β for ScV_6Sn_6 , obtained from the fitting of the magnetoresistance curves using the polynomial function: $\Delta\rho/\rho_{H=0} = \alpha$

$+ \beta(\mu_0 H)^{\eta}$. Vertical grey line marks the charge order temperature with $T^* \approx 80$ K.

1.4 Reviewer's comment: Concerning Fig.2d, the color bar appearing there has to be corrected. Moreover, there are some features of the ARPES map that appear (or, at least, are clearly visible) just in the upper half of the first Brillouin zone, especially in the region closer to the Gamma point. Can the authors explain the origin of this asymmetry? Can it be regarded as a signature of nematicity?

Our response: We thank the reviewer for the valuable suggestion and interesting point. The color bar has been corrected in the revised manuscript. We agree with the referee on that some features of the ARPES map are more clearly visible in the upper half of the first Brillouin zone, especially in the region closer to Γ . This asymmetry can be generally attributed to the suppression of spectral weight in most ARPES measurements, instead of the signature of nematicity.

As shown in Fig. R2, in spite of relatively weak intensity, it can still be captured in our ARPES measurements (marked by green arrows), which indicates that the sixfold symmetry is preserved. Moreover, in the second Brillouin zone, the same features with equal intensity exist in both the upper and lower halves of the Brillouin zone, as shown by the red arrows in Fig. R2. Therefore, this asymmetry is unlikely the signature of nematicity. In conclusion, the asymmetric intensity of the band structure in the first Brillouin zone is not likely related to the nematicity, and can be explained by matrix element effects. Whether the nematicity exists in ScV_6Sn_6 is beyond the scope of our manuscript and requires further investigation.

Figure R2: Fermi surface measured by ARPES.

1.5 Reviewer's comment: *At the end of the first column of page 6 of the manuscript, the authors state: "So far, no theoretical proposal for the orbital current order was reported for ScV6Sn6." However, I am not sure this is the case. Indeed, in [Grandi et al., Phys. Rev. B 107, 155131 (2023)], a state that hosts orbital currents is also suggested for the $\sqrt{3}\times\sqrt{3}$ reconstruction of the kagome lattice, even if only a phenomenological description of the state is provided. As a consequence, I think the quoted sentence has to be modified.*

Our response: We agree with the Reviewer with this important point. We are now aware of the wonderful theoretical work by Grandi et. al., and modified the corresponding sentences in the discussion section.

1.6 Reviewer's comment: *I suggest you to describe, maybe in the supplemental material, how the fit shown in Fig.4a is performed for the sake of reproducibility.*

Our response: Following the suggestion of the Reviewer, we included the details of high-field data analysis in the supplementary information.

1.7 Reviewer's comment: *If I understand Fig.4b correctly, the error bars have almost the same size as the symbols. This makes them difficult to be seen, and for this reason, I suggest reducing the size of the symbols (same in Fig.3e).*

Our response: Following the suggestion of the Reviewer, we reduced the size of the symbols in Figure 4b as well as in Figs. 3b-e.

Reply to the Reviewer 2:

2.1 Reviewer's comment: *Kagome physics is in the frontier of condensed matter research. The authors found enhanced magnetic response in single crystal ScV_6Sn_6 below the charge order temperature $T^* \sim 80$ K using ZF- μSR and high-field μSR measurements, indicating the time-reversal symmetry-breaking in kagome paramagnet ScV_6Sn_6 . However, I cannot recommend its publication in Nature Communications at the present stage, there are some comments on this work need to be addressed.*

Our response: We are thankful to the Reviewer for carefully reading the manuscript and providing constructive and useful comments. We considered the suggestions from the Reviewer in the revised version.

2.2 Reviewer's comment: *The specific heat capacity curve shows a peak around 80 K, indicating the charge order transition. Is there the same response in the magnetic susceptibility and resistivity curves? Moreover, the transition temperature is lower than that ~ 90 K in previous work, is this related to the quality of sample?*

Our response: Both the magnetization and the resistivity shows a clear anomaly across the charge order transition temperature $T^* \approx 80$ K in ScV_6Sn_6 , as shown in Figures R3a and b, respectively.

Figure R3: The temperature dependence of magnetization and resistivity for the sample with $T^* \approx 80$ K.

The high-quality of the crystal was assessed using single-crystal X-ray diffraction (crystal structure) and X-ray fluorescence (composition). The X-ray single-crystal diffraction measurement were done using hard X-ray source (AgK alfa, $\lambda = 0.56\text{\AA}$) to mitigate the effect of absorption. The whole sphere of reflection was measured down to very good resolution in direct space (0.5 \AA) to disentangle all possible features of the crystal structure. We detect no impurity elements; the crystal structure is ordered and stoichiometric according to the results of the refinement. We checked possible deviations from the stoichiometry by refining occupancies; they all appeared to be within 3σ from unity. We also checked that the difference

Fourier maps are featureless. In sum, minor deviation of properties between samples with $T^* \simeq 80$ K and $T^* \simeq 90$ K samples could stem from minor differences in microstructure undetectable using the employed probes. What we can say based on the X-ray diffraction measurements that the sample with $T^* \simeq 90$ K is less stoichiometric than the sample with $T^* \simeq 80$ K. This is the reason why we performed muon-spin rotation experiments on the $T^* \simeq 80$ K sample.

To further elaborate on the question of the Reviewer, we carried out magnetoresistance measurements for the $T^* \simeq 90$ K sample and the results are shown in Figure R4 and R5. Absolute value (see Figure R4) as well as the shape of $\rho(H)$ at various temperatures (see Figures R5a and b) is very identical for the two samples. This means that despite the differences in T^* , the physics of low-temperature charge ordered state remains eventually the same for both samples.

Figure R4: The magnetoresistance for the sample with $T^* \simeq 90$ K, measured at various temperatures above and below the charge ordering temperature.

Figure R5: The temperature dependence of the parameter n for the samples with $T^* \approx 80$ K (a) and $T^* \approx 90$ K (b), obtained from the fitting of the magnetoresistance curves using the polynomial function: $\Delta\rho/\rho_{H=0} = \alpha + \beta(\mu_0 H)^n$.

2.3 Reviewer's comment: *The data in Fig.3 (d)-(e) are obtained from two sets of detectors, can they describe more in detail? It is noticed that the rate Γ_{34} increases about $0.03 \mu\text{s}^{-1}$ below T^* in Fig.3 (d), but it shows almost temperature independent in the whole temperature range in Fig.3 (e).*

Our response: We thank the reviewer for this interesting question. To address this question, in Fig. R6 we show the schematic illustration of the muon spin precession around the internal magnetic field, for two extreme cases: the local internal field B_{int} is perpendicular to c and B_{int} parallel to c . For the internal field direction, shown in the top panel of Fig. R6, the μSR signal from 1-2 (F-B) detectors exhibits the maximum amplitude and no oscillations will be detected in the 3-4 (L-R) detectors. The opposite will be observed for the configuration shown in the bottom panel of Fig. R6. Thus, by evaluating the data from all four detectors one can obtain useful information on the direction of the internal field.

Since we see enhanced electronic relaxation below T^* in both Δ_{12} and Δ_{34} we conclude that the local field at the muon site cannot lie purely along the c-axis direction (this would lead to an absence of the term Δ_{34}). However, any orientation of the local field which has a significant component in the basal plane is consistent with our data. The corresponding statement was added in the revised manuscript.

Regarding Γ_{34} , it indeed shows an increase below T^* , which is clearly demonstrated in Fig. 3d. In Fig. 3e, we intentionally kept both the Γ_{34} and Γ_{12} constant as a function of temperature. Therefore, we make the following statement in the caption of Figure 3: In panel e the rates Γ_{34} and Γ_{12} are kept constant as a function of temperature.

Figure R6: (a) A schematic overview of the experimental setup for the muon spin forming 45°

with respect to the c- axis of the crystal. The sample was surrounded by four detectors: Forward (F), Backward (B), Left (L) and Right (R). (b-c) Schematic illustration of the muon spin precession around the internal magnetic field for two cases: (b) The field is perpendicular to the c-axis and points towards the L-detector. θ is the angle between the magnetic field and the muon spin polarization at $t = 0$. (c) The field is parallel to the c-axis of the crystal and points towards the F-detector.

2.4 Reviewer's comment: *The relaxation rate under high fields shows clear increase towards low temperatures. Around T^* , it shows different behavior (non-monotonous or monotonous) under various fields, can they explain it?*

Our response: To begin with, we recall that Gaussian relaxation rate Δ_{34} shows a non-monotonous temperature dependence; namely, a peak coinciding with the onset of the charge order, which decreases to a broad minimum before increasing again towards lower temperatures. Δ_{12} instead shows a weak minimum at T^* with the significant increase at lower temperatures. The Gaussian component includes the field distribution at the muon site created by a dense network of weak electronic moments and plus temperature independent contribution from nuclear moments. The onset of charge order might alter the electric field gradient experienced by the nuclei, due to the fact that the quantization axis for the nuclear moments depends on the electric field gradients (EFG), and correspondingly the magnetic dipolar coupling of the muon to the nuclei. This can induce a change in the nuclear dipole contribution to the zero-field μ SR signal and may explain the small maximum or minimum in Δ_{34} and Δ_{12} , respectively, at the onset of T^* . However, the significant increase of both Δ_{34} and Δ_{12} at lower temperatures is not related to change of the EFG and shows a considerable contribution of electronic origin (dense moments) to the muon spin relaxation in the charge ordered state.

Reviewer is right that a non-monotonous behavior of the relaxation rate is also seen in the μ SR data, measured in low magnetic fields of 0.01T (see Fig. 4b of the manuscript), applied parallel to the c-axis. This shows that the effect of nuclear contribution on the relaxation rate at the charge order transition is still visible in low fields. However, at higher fields such as 2T, 4T, 6T and 8T, the rate does not show decrease of the rate T^* but rather shows a clear and monotonous increase with decreasing temperature. As the nuclear contribution to the relaxation cannot be enhanced by an external field, this indicates that the low-temperature relaxation rate in magnetic fields higher than 0.01T is dominated by the electronic contribution and minor effect of nuclear contribution at the transition is diminished. Remarkably, we find a factor of six enhancement of the relaxation rate with the onset of T^* in 8T compared to zero-field, showing a strong field-induced enhancement of the electronic response.

2.5 Reviewer's comment: *The authors compared some similarities and differences between ScV_6Sn_6 and AV_3Sb_5 , what is the novelty of this work?*

Our response: The series of compounds AV_3Sb_5 ($A = K, Rb, Cs$) form the first kagome-based family that exhibit a cascade of symmetry-broken electronic orders, including charge order and superconductivity. An important feature of charge order, which was reported for all three compounds and that has been intriguing scientists over the past two years, is the breaking of time-reversal symmetry. Recently, charge order was reported in non-superconducting ScV_6Sn_6

that has a similar vanadium structural motif as the AV_3Sb_5 compounds. However, the time-reversal symmetry-breaking nature of charge order in ScV_6Sn_6 remained elusive and unresolved. We provide the first microscopic study of the magnetic fingerprints in the charge ordered phase of ScV_6Sn_6 with vanadium kagome lattice. Our approach is based on combining zero-field and high-field muon-spin rotation methods as well as magneto-transport measurements, which provide a sensitive way to identify weak electronic response of charge order. Our results are indicative of a time-reversal symmetry breaking charge order in ScV_6Sn_6 and provide fresh insights into the nature of the charge ordered state. Taken together with the hidden magnetism found in AV_3Sb_5 ($A = K, Rb, Cs$) and FeGe kagome systems, our results suggest ubiquitous time-reversal symmetry-breaking in charge ordered kagome lattices. Compared to the AV_3Sb_5 and FeGe compounds, the $HfFe_6Ge_6$ -type compounds offer improved tunability making them an ideal platform to explore the curious CDWs in transition metal kagome systems. Therefore, the finding presented in this work is significant, and of relevance to the broad readership of Nature Communications.

After we posted our work about TRS breaking charge order on condmat, several papers appeared, reporting the anomalous Hall effect in charge ordered state and anomalous normal state transport behaviour, which support our results. These interesting transport papers are now cited in the revised version of our paper.

2.6 Reviewer's comment: *There are some errors in the manuscript. For example, the labels (b)-(d) in Fig.3 are unclear. In addition, the color of the fitting line in Fig.4 (a) is blue, not black.*

Our response: We are grateful to the reviewer for pointing out this unintentional typographical issue. It is corrected in the revised version.

Reply to the Reviewer 3:

3.1 Reviewer's comment: *The manuscript reports on very interesting muon spin spectroscopy findings in a new Kagome compound with striking similarities to those obtained by some of the authors on the very actively studied AV3Sb5 (135) series. It is very clearly written and it deserves consideration for Nature Communications in view of the many order parameters that these structures may host, and of their possible interplay. The magnetic nature of the low temperature muon relaxation is proven beyond doubt by the zero field (ZF), longitudinal field (LF) and high transverse field (HTF) data of Fig. 3 and 4. However a number of points related to data interpretation need addressing:*

Our response: We are thankful to the Reviewer for supporting its consideration in Nature communications and for appreciating our work, highlighted by encouraging phrases. We also thank the Reviewer for providing constructive comments.

3.2 Reviewer's comment: *An anomaly appears in the ZF Kubo-Toyabe (KT) static widths at the known onset of the CDW. This is attributed to the change in the EFG across the onset. However, a change in EFG could give rise to an anisotropic (almost) step-wise change in the relaxation, reflecting the temperature dependence of the CDW order parameter, i.e. widening towards low temperatures. The bump showing up in the 34 and not in the 12 pair of detectors is hard to reconcile with this interpretation, since it completely disappears below 50 K where the two relaxation rates overlap again.*

Our response: We thank the Reviewer for pointing this out and agree with the Reviewer. In the manuscript, we only discuss the possibility of change in EFG across the charge order transition as a potential source of change in zero-field relaxation rate at the charge order transition. However, as we describe from high field μ SR experiments, this appears to be only a minor contribution to the observed temperature dependence, as the relaxation rate strongly enhanced by the magnetic fields. In addition, the observed μ SR relaxation rate does not resemble the expected T-dependence of the charge order parameter. Following the suggestion of the reviewer, we include the argumentation from the Reviewer in the revised manuscript.

3.3 Reviewer's comment: *"At higher fields such as 2 T, 4 T, 6 T, and 8 T, the rate shows a clear and stronger increase towards low temperatures within the charge ordered state." This statement implies that the magnetic origin of the rate correlates with T^* , but the correlation is much weaker than implied. In particular HTF data of Fig. 4b show a smooth continuous decrease across T^* . It seems compatible with a slowing down mechanism connected to a paramagnetic susceptibility of Curie-Weiss type, i.e. with a finite exchange interaction. This would be still non trivial in a metal with CDW instabilities, but it must not necessarily imply orbital currents developing below the CDW transition.*

Our response: According to our longitudinal field experiments (see Fig. 3c of the manuscript) the relaxation is due to spontaneous fields which are static on the microsecond timescale. Therefore, we can discard the slow (para-)magnetic fluctuations as a source of the muon spin relaxation.

3.4 Reviewer's comment: *"We also note that a weak but non-negligible field effect on the relaxation rate is observed above T^* within a 20-30 K temperature range, which may point towards the charge density wave fluctuations preceding the phase transition." CDW fluctuations are very unlikely the cause of a relaxation that seamlessly reconnects to magnetic fluctuations to produce a continuous rate variation across T^* . It is suggested that this mention is either expanded and justified fully, or dropped.*

Our response: We are grateful to the Reviewer for raising this important issue. What we meant is the short-range charge order rather than fluctuations. We made a corresponding change in the manuscript and the revised sentence reads as the following: *"We also note that a weak but non-negligible field effect on the relaxation rate is observed above T^* within a 20-30 K temperature range, which may point towards **the short-range charge order** preceding the phase transition."* Short-range charge order was also suggested by inelastic X-ray scattering and magneto-transport measurements.

3.5 Reviewer's comment: *There is no reference in the text to the temperature dependence of the shift for the sample component of the HTF data. It would be interesting to compare this shift with the spin susceptibility, with temperature as an implicit parameter (Clogston-Jaccarino plot). The hyperfine coupling could be obtained this way and it could provide further insight. In any case, shifts and susceptibility are mandatory (in the main text or as supplemental information). They clarify whether the sample displays any macroscopic magnetic behavior at all, and whether magnetic impurities are present in the sample.*

Our response: In general, the Knight shift is due to the paramagnetism of the host material, and is therefore closely related to its bulk susceptibility χ . In some simple cases, χ and the Knight shift K are linearly related: $K = A\chi$, where A is a coupling constant. Then if χ depends on temperature, a plot of $K(T)$ versus $\chi(T)$, with temperature T as implicit parameter (the so-called Clogston-Jaccarino plot), is a straight line with zero intercept.

In Figure R7 of this response, we show the temperature dependence of the Knight shift K_{exp} (local susceptibility) for ScV_6Sn_6 , measured under the c -axis magnetic fields of $\mu_0H = 2$ T, 4 T, 6 T, and 8 T and the temperature dependence of the macroscopic magnetization, measured in the c -axis magnetic field of 2 T. As it is clear from Figure R7, both the local susceptibility and the macroscopic susceptibility shows decrease at the charge order transition temperature $T^* \simeq 80$ K, followed by an increase at lower temperatures. However, the increase of K_{exp} occurs below 30 K while magnetization shows an increase below 70 K. So, while overall temperature dependence between K_{exp} and M looks qualitatively very similar, quantitatively there is a breakdown of the proportionality of the μ^+ Knight shift to the measured bulk susceptibility.

The Knight shift K_{exp} , shown in Figure R7, is an experimental Knight shift. Due to the irregular shape of the crystals, it is not possible to consider the demagnetization factor and estimate precise magnitude of the Knight shift and its temperature dependence. Therefore, we can not further discuss the comparison between local and bulk susceptibilities. Following the comment of the reviewer we show the figure R7 along with the corresponding text in the supplementary information.

Figure R7: (Left axis) The temperature dependence of the Knight shift K_{exp} (local susceptibility) for ScV_6Sn_6 , measured under the c-axis magnetic fields of $\mu_0H = 2$ T, 4 T, 6 T, and 8 T. (Right axis) The temperature dependence of the macroscopic magnetization, measured in the c-axis magnetic field of 2 T.

3.6 Reviewer's comment: *"The exponential term Γ may e.g. be due to the presence of electric field gradients, causing deviations from the GKT-like spectrum or dilute electronic moments. [...] There is also a note-worthy increase in the relaxation rate Γ 34 upon lowering the temperature below the charge ordering temperature T^* ...". The Γ parameter belongs to a factor of the model function(1), i.e. it is assumed to originate from an uncorrelated mechanism. If the rate were temperature-independent it could account for an indirect EFG effect. However, the low temperature increase is irreconcilable with a CDW order parameter variation (even less, fluctuation), so the EFG explanation is shaky.*

Our response: We agree with the reviewer on this point. we only discuss the possibility of change in EFG across the charge order transition as a potential source of change in zero-field relaxation rate at the charge order transition. However, as we describe from high field μSR experiments, this appears to be only a minor contribution to the observed temperature dependence, as the relaxation rate strongly enhanced by the magnetic fields.

In the revised manuscript, we removed *"The exponential term Γ may e.g. be due to the presence of electric field gradients, causing deviations from the GKT-like spectrum"* and the now text reads as the following: *"The exponential term Γ is due to the dilute electronic moments. Δ_{34} shows a non-monotonous temperature dependence; namely, a peak coinciding with the onset of the charge order, which decreases to a broad minimum before increasing again towards lower temperatures. Δ_{12} instead shows a weak minimum at T^* with the significant increase at lower temperatures. The onset of charge order might alter the electric field gradient (EFG) experienced by the nuclei, due to the fact that the quantization axis for the nuclear moments depends on the electric field gradients, and correspondingly the magnetic dipolar*

coupling of the muon to the nuclei. This can induce a change in the nuclear dipole contribution to the zero-field μ SR signal and may explain the small maximum or minimum in Δ_{34} and Δ_{12} , respectively, at the onset of T^* . However, the significant increase of both Δ_{34} and Δ_{12} at lower temperatures is difficult to explain with the change of the EFG and suggests a considerable contribution of electronic origin (dense moments) to the muon spin relaxation in the charge ordered state. There is also a noteworthy increase in the relaxation rate Γ_{34} upon lowering the temperature below the charge ordering temperature T^* , which is better visible in the inset of Fig.~3d. Moreover, our high field μ SR results presented below definitively prove that there is indeed a strong contribution of electronic origin to the muon spin relaxation below the charge ordering temperature.’’

3.7 Reviewer’s comment: While the investigation of the magnetic properties of these compounds is of utmost importance, the comparison with the 135 materials where ‘‘Muons couple to the closed current orbits below T^* , leading to an enhanced internal field width sensed by the muon ensemble concurrent with the charge order’’ is less evident from the experimental data presented in the manuscript. This has been partially discussed in the previous points, but it is also visible in the different trends of the relaxation rate observed in ZF and applied field that are seemingly uncorrelated with T^* (actually this is the case also for the ZF measurements of the 135, where an upturn in the relaxation rate happens below the CDW transition). In present case however one observes temperature dependent phenomena starting well below (~ 50 K in ZF) and well above (> 100 K for $TF > 4$ T) the critical temperature of the CDW. Can the author comment on this point and provide further evidence of their statement?

Our response: We kindly disagree with the Reviewer with the following statement: ‘‘In present case however one observes temperature dependent phenomena starting well below (~ 50 K in ZF).’’ As it is shown in Figure 3d, the exponential relaxation rate Γ_{34} shows the increase right below $T^* \approx 80$ K. Since Γ_{34} is purely electronic in origin, it is expected to show the true onset of magnetism in zero-field. The Gaussian rate Δ_{12} also shows an increase with the onset of $T^* \approx 80$ K. But the relaxation rate Δ_{34} shows a peak coinciding with the onset of the charge order, which decreases to a broad minimum before increasing again towards lower temperatures. The Gaussian component includes the field distribution at the muon site created by a dense network of weak electronic moments and plus temperature independent contribution from nuclear moments. The onset of charge order may modify the magnetic dipolar coupling of the muon to the nuclei. This can induce a change in the nuclear dipole contribution to the zero-field μ SR signal and may explain the small maximum in Δ_{34} around T^* , leading to observed non-monotonous behaviour. Due to the presence of both electronic and nuclear effects in Δ_{34} , it can not be used to determine true onset of magnetic response.

A non-monotonous behavior of the relaxation rate is also seen in the μ SR data, measured in low magnetic fields of 0.01T (see Fig. 4b of the manuscript), applied parallel to the c-axis. This shows that the effect of nuclear contribution on the relaxation rate at the charge order transition is still visible in low fields. However, at the field of 2T the rate does not show decrease of the rate below T^* but rather shows a clear and monotonous increase with decreasing temperature with the onset of T^* . As the nuclear contribution to the relaxation cannot be enhanced by an external field, this indicates that the low-temperature relaxation rate in magnetic fields higher than 0.01T is dominated by the electronic contribution and minor effect of nuclear contribution at the transition is diminished. Under the applied fields of 4T, 6T and 8T, transition becomes

smear out and it is difficult to determine the onset of magnetic response. A weak field effect on the relaxation rate, observed above T^* , may point towards the short-range charge which becomes more pronounced or visible under high fields." Short-range charge order was suggested by inelastic X-ray scattering and magneto-transport measurements.

Regarding the 135 materials, in all three compounds (K,Rb,Cs)V₃Sb₅ the onset of magnetic response coincides with the onset of charge order, as it was concluded based on combination of zero-field and high-field μ SR experiments. But in RbV₃Sb₅ and CsV₃Sb₅, we see two-step increase of the electronic relaxation rate Γ . For instance, in RbV₃Sb₅, the electronic response consists of a noticeable enhancement at $T_1^* \simeq 110$ K, which corresponds to the charge-order transition temperature T_{co} , and a stronger increase below $T_2^* \simeq 50$ K. From the measurements on single crystals, we concluded that below $T_1^* \simeq 110$ K the internal field lies mostly within the ab -plane of the crystal, while below $T_2^* \simeq 50$ K the internal field also acquires an out-of-plane component. The lower-temperature increase of the relaxation rate at $T_2^* \simeq 50$ K is suggestive of another ordered state that modifies magnetic response. An obvious candidate is a secondary charge-ordered state onsetting at $T_2^* \simeq 50$ K. Indeed, experimentally, it has been reported that RbV₃Sb₅ and CsV₃Sb₅ kagome metals may display two charge-order transitions. Theoretically, different charge-order instabilities have been found in close proximity.

TRS breaking charge order in AV₃Sb₅ was interpreted in terms of orbital current order. We note that drastic magnetic-field-induced chiral current order was also reported theoretically which lines up with our high field μ SR results. Unconventional charge order and orbital current order has also been proposed for ScV₆Sn₆ based on Ginzburg-Landau and mean-field analysis. According to the theoretical modelling, there is extremely small net flux and thus the small net magnetic moment in the unit cell of the order. The suggested orbital current was reported to be homogeneous on the lattice, however alternating in its flow, which would produce inhomogeneous fields at the muon site. Within this framework, muons may couple to the closed current orbits below T^* , leading to an enhanced internal field width sensed by the muon ensemble concurrent with the charge order. Despite the fact that μ SR results seems to be compatible with the picture of orbital current order, we cannot conclude on the microscopic origin of the TRS breaking field in this system. But, our results provide key evidence that the magnetic and charge channels of ScV₆Sn₆ are strongly intertwined, which can give rise to complex and collective phenomena. This is an experimental finding which stands even without the knowledge of its microscopic origin.

Following the comments of the Reviewer, for clarity we added several sentences the revised manuscript.

3.8 Reviewer's comment: *Minor point: Fig 4a caption does not specify the short-time-window apodization function used in the FT amplitude plot, and indeed adding a longer time window function FT plot could show the narrow nature of the high frequency peak.*

Our response: We specify this point in the caption of Fig. 4a of the revised manuscript.

3.9 Reviewer's comment: *Typos: * TRS is not defined. * Correct the angel in "MuSR methods".*

Our response: We are thankful to the Reviewer for carefully reading the manuscript and for pointing out this unintentional typographical issue. This is corrected in the revised manuscript.

Reply to the Reviewer 4:

4.1 Reviewer's comment: *I co-reviewed this manuscript with one of the reviewers who provided the listed reports. This is part of the Nature Communications initiative to facilitate training in peer review and to provide appropriate recognition for Early Career Researchers who co-review manuscripts.*

Our response: We very much appreciate the time and effort put forward by all reviewers involved in the review process.

REVIEWER COMMENTS

Reviewer #1 (Remarks to the Author):

The authors properly addressed all the points I raised and, in my opinion, the comments of the other referees. I have only a few remarks:

1) In the Supplementary Information, the reference to “Figure S2a and a” and the repetition “show show” should be modified in Sec.III. Also, the reference to “ $\chi(H)$ ” needs to be amended.

2) In Sec.IV of the Supplementary Information, there is a wrong reference to Figure R7, which should be Figure S5. In the same section, “magnetiyation” has to be corrected.

3) In Sec. VI of the Supplementary Information, below Eq.(1), you should remove the temporal dependence of P_S and P_{BG} from the subscript. Below Eq.(3), you should correct the repetition “from the from the”.

Reviewer #3 (Remarks to the Author):

I have read the revised manuscript together with the responses of the authors to my comments and the ones of the other Referees. The authors have submitted a manuscript similar to the original version but they have extended the supplemental material. I appreciate the additional work and the results, and have re-read my original report.

I still have some comments concerning the discussion of the results. Let me start by supporting their view and confirm that, given the size of the change of the relaxation rate, a TRBS ground state is very likely.

Nonetheless, this conclusion is supported with an analysis that lacks important considerations that still need to be clarified. This would also allow the reader to appreciate the complexity of the picture emerging from the measurements.

I report here the relevant part of the reply from the authors, in order to provide accurate and specific observations.

- > In the revised manuscript, we removed “The exponential term Γ may e.g. be due to the
- > presence of electric field gradients, causing deviations from the GKT-like spectrum”

This is not what I intended and it is an unfortunate change. If one considers a static muon interacting only with (static) nuclei, the deviation from the GKT is due to three contributions: one is the EFG, the second one is the effect induced by the muon on the polarization of the neighboring atoms and the third one is due to quantum effects. Surprisingly the second one can be approximated also with classical simulations (see Physics Letters A 162 206 (1992)). All these contributions are temperature independent if and only if the nuclei don't move and the electronic charge distribution is preserved in the temperature interval of interest. Since Γ includes the three contributions described above (in addition, possibly, to other ones of electronic origin), its temperature evolution is a complex mixture of the temperature evolution of different interactions.

- > [...] now text reads as the following: “The exponential term Γ is due to the dilute electronic
- > moments.

This is not correct, or at least the data do not allow to conclude that electronic moments are the only contributors. If the authors do believe this is the case, they should show how they reach this conclusion.

- > Δ_{34} shows a non-monotonous temperature dependence; namely, a peak coinciding
- > with the onset of the charge order, which decreases to a broad minimum before increasing again
- > towards lower temperatures. Δ_{12} instead shows a weak minimum at T^* with the significant
- > increase at lower temperatures. The onset of charge order might alter the electric field gradient
- > (EFG) experienced by the nuclei, due to the fact that the quantization axis for the nuclear
- > moments depends on the electric field gradients, and correspondingly the magnetic dipolar
- > coupling of the muon to the nuclei. This can induce a change in the nuclear dipole contribution
- > to the zero-field μ SR signal and may explain the small maximum or minimum in Δ_{34} and Δ_{12} ,
- > respectively, at the onset of T^* . However, the significant increase of both Δ_{34} and Δ_{12} at lower
- > temperatures is difficult to explain with the change of the EFG and suggests a considerable
- > contribution of electronic origin (dense moments) to the muon spin relaxation in the charge
- > ordered state.

This is indeed the most likely explanation, but EFG is not the only ingredient.

There are other options such as an additional modulation of the lattice structure that slightly alters the nuclear positions around the muon. I believe the authors can easily dismiss this. Indeed a rough order of magnitude estimate based on second moments estimates of muon's position indicates that a perturbation of the order of 0.1 Angstrom for the atom(s) closest(s) to the muon should take place. This is a large effect that is not seen by any other technique. Moreover, this lattice distortion should be varying in temperature with a rather unusual trend.

Nonetheless, from Δ_{34} in fig 3e a strong departure is only seen below 40K. The authors comment on Γ_{34} of fig. 3D, but, as discussed above, this contribution is linked to the ones building up sigmas. The then focus on the high field results stating that:

- > There is also a noteworthy increase in the relaxation rate Γ_{34} upon lowering the
- > temperature below the charge ordering temperature T^* , which is better visible in the inset of
- > Fig.~3d. Moreover, our high field μ SR results presented below definitively prove that there is
- > indeed a strong contribution of electronic origin to the muon spin relaxation below the charge
- > ordering temperature."

Now moving to the next point in the rebuttal letter (3.7):

- > As it is shown in Figure 3d, the exponential relaxation rate Γ_{34} shows the increase
- > right below $T^* \simeq 80$ K. Since Γ_{34} is purely electronic in origin,

This last sentence is not true (see above) and the authors did point it out correctly in the first version of the manuscript.

- > it is expected to show the true
- > onset of magnetism in zero-field. The Gaussian rate Δ_{12} also shows an increase with the onset
- > of $T^* \simeq 80$ K. But the relaxation rate Δ_{34} shows a peak coinciding with the onset of the charge
- > order, which decreases to a broad minimum before increasing again towards lower

> temperatures. The Gaussian component includes the field distribution at the muon site created
> by a dense network of weak electronic moments and plus temperature independent contribution
> from nuclear moments. The onset of charge order may modify the magnetic dipolar coupling
> of the muon to the nuclei. This can induce a change in the nuclear dipole contribution to the
> zero-field μ SR signal and may explain the small maximum in $\Delta 34$ around T^* , leading to
> observed non-monotonous behaviour. Due to the presence of both electronic and nuclear
> effects in $\Delta 34$, it can not be used to determine true onset of magnetic response.

This is absolutely correct and, in light of the above discussion, it is valid also for Γ .

> A non-monotonous behavior of the relaxation rate is also seen in the μ SR data, measured in
> low magnetic fields of 0.01T (see Fig. 4b of the manuscript), applied parallel to the c-axis. This
> shows that the effect of nuclear contribution on the relaxation rate at the charge order transition
> is still visible in low fields. However, at the field of 2T the rate does not show decrease of the
> rate below T^* but rather shows a clear and monotonous increase with decreasing temperature
> with the onset of T^* . As the nuclear contribution to the relaxation cannot be enhanced by an
> external field, this indicates that the low-temperature relaxation rate in magnetic fields higher
> than 0.01T is dominated by the electronic contribution and minor effect of nuclear contribution
> at the transition is diminished. Under the applied fields of 4T, 6T and 8T, transition becomes
smeared out.

I totally agree with the authors and this leaves three measurements showing two different trends: ZF and 0.01 T that have apparently the same behavior and the trend acquired at 2 T, which is markedly different. In the remaining acquisitions as a function of the applied field, a clear transition at T^* is not visible (I agree with the last sentence reported above).

Nonetheless and surprisingly, the Knight shift measurements (fig. R7) clearly show a first deviation at T^* and a second one at about 25K which is exactly the temperature where an upturn is visible also in Fig 3e and, in my eyes, also in the 0.01 T measurements of fig 4b. This suggests that the onset temperature for the additional relaxation in zero and low fields is distinct from the onset of charge order, despite a single transition being visible in fig 4b.

In conclusion, my comments are:

1. the description of the physical origin of Γ should be clarified and,
2. taken together, the simplest interpretation of the Knight Shift measurements and the ZF, and TF measurements at 0.01 T and 2 T is that two characteristic temperatures can be identified (one being the well known CDW transition, of course). Do the authors agree on this point? While one can certainly leave the microscopic interpretation to future works, two temperatures potentially means two different order parameters, which is an important information for theories of electronic states in kagome materials.

Reviewer #4 (Remarks to the Author):

“Hidden magnetism uncovered in charge ordered bilayer kagome material ScV_6Sn_6 (NCOMMS-23-18321A)”.’.

Reply to the Reviewer 1:

1.1 Reviewer’s comment: *The authors properly addressed all the points I raised and, in my opinion, the comments of the other referees. I have only a few remarks:*

Our response: We are very grateful to the reviewer for supporting our manuscript for publication in Nature Communications. We are thankful to the Reviewer for carefully reading the manuscript and for pointing out the typographical issues. This is corrected in the revised manuscript.

1.2 Reviewer’s comment: *In the Supplementary Information, the reference to “Figure S2a and a” and the repetition “show show” should be modified in Sec.III. Also, the reference to “ H ” needs to be amended.*

Our response: This is corrected in the revised manuscript.

1.3 Reviewer’s comment: *In Sec.IV of the Supplementary Information, there is a wrong reference to Figure R7, which should be Figure S5. In the same section, “magnetiyation” has to be corrected.*

Our response: This is corrected in the revised manuscript.

1.4 Reviewer’s comment: *In Sec. VI of the Supplementary Information, below Eq.(1), you should remove the temporal dependence of P_S and P_{BG} from the subscript. Below Eq.(3), you should correct the repetition “from the from the”.*

Our response: This is corrected in the revised manuscript.

Reply to the Reviewer 3: *I have read the revised manuscript together with the responses of the authors to my comments and the ones of the other Referees. The authors have submitted a manuscript similar to the original version but they have extended the supplemental material. I appreciate the additional work and the results, and have re-read my original report.*

1.1 *I still have some comments concerning the discussion of the results. Let me start by supporting their view and confirm that, given the size of the change of the relaxation rate, a TRBS ground state is very likely. Nonetheless, this conclusion is supported with an analysis that lacks important considerations that still need to be clarified. This would also allow the reader to appreciate the complexity of the picture emerging from the measurements.*

Our response: We are very grateful to the reviewer for supporting our conclusion about time-reversal symmetry-breaking charge ordered ground state in ScV_6Sn_6 . Reviewer had two main comments which we considered fully in the revised version.

In conclusion, my comments are:

1.2 *The description of the physical origin of Γ should be clarified.*

Our response: We thank the reviewer for this important point. Following the suggestion of the Reviewer, we extended the discussion on the physical origin of the exponential relaxation rate Γ . In the revised version, we clearly state that the deviation from a purely GKT like spectrum which is accounted for by the exponential term Γ may e.g. originate from the entanglement of the muon with neighboring quadrupolar nuclei, modification of the nuclear positions around the muon due to charge order, or slowly fluctuating dilute electronic moments. We also mention that we can dismiss the structural distortion or the change in EFG being origin for the increase of the relaxation rate due to the following reasons:

- (1) Change of EFG across the charge order temperature can induce a change in the nuclear dipole contribution to the zero-field μSR signal and may explain the small maximum or minimum in Δ_{34} and Δ_{12} , respectively, at the onset of T^* . However, the significant increase of both Δ_{34} and Δ_{12} at lower temperatures is difficult to explain with the change of the EFG and suggests a considerable contribution of electronic origin (dense moments) to the muon spin relaxation in the charge ordered state.
- (2) A rough order of magnitude estimate yields that the structural distortions of the order of 0.1 Angstrom for the atoms closest to the muon would be needed to explain the observed effect in the second moment of the measured field distribution. This is a large effect that has not been seen by any other technique. Moreover, this lattice distortion should be varying in temperature with a rather unconventional trend.
- (3) Most importantly, our high field μSR results (field-induced enhancement of the relaxation rate) definitively prove that there is indeed a strong contribution of electronic origin to the muon spin relaxation below the charge ordering temperature.

1.3 *Taken together, the simplest interpretation of the Knight Shift measurements and the ZF, and TF measurements at 0.01 T and 2 T is that two characteristic temperatures can be identified (one being the well known CDW transition, of course). Do the authors agree on this point? While one can certainly leave the microscopic interpretation to future works, two temperatures potentially means two different order parameters, which is an important information for theories of electronic states in kagome materials.*

Our response: We agree with the reviewer that zero-field and low field relaxation rate as well as Knight-shift measurements indicate two characteristic temperatures 25 K and $T^* = 80$ K, which may point towards two different order parameters. Following the comment of the reviewer, we mention this point in two different places of the manuscript. Since we do not have additional measurements showing the transition at 25 K and also do not understand the microscopic origin of it, we prefer not to put too much emphasis on it.

Reply to the Reviewer 4: *I co-reviewed this manuscript with one of the reviewers who provided the listed reports. This is part of the Nature Communications initiative to facilitate training in peer review and to provide appropriate recognition for Early Career Researchers who co-review manuscripts.*

Our response: We very much appreciate the time and effort put forward by all reviewers involved in the review process.